# Uncertainty and Emerging Tensions in Organizational Change: A Grounded Theory Study on the Orchestrating Role of the Change Leader

Francesco Virili [1,*] and Cristiano Ghiringhelli [2]

1 Department of Economics and Business, University of Sassari, Via Muroni 25, 07100 Sassari, Italy
2 Department of Human Sciences for Education, University of Milan-Bicocca, Piazza Ateneo Nuovo 1, 20126 Milano, Italy; cristiano.ghiringhelli@unimib.it
* Correspondence: fvirili@uniss.it

**Abstract:** We draw on the grounded theory methodology to analyze an automation project in a global parcel delivery company, as implemented in three parcel sorting hubs in two countries, seeking to identify key factors in successful change and the role of the change leader. We show that a crucial—but often neglected—aspect of successful change is the detection and management of tensions stemming from uncertainty. By recognizing and managing uncertainty and tensions (in this case, manual vs. automated, corporate vs. site, and planned vs. emergent), the change leader, here the industrial engineering function, can orchestrate the differing views and expectations of corporate actors toward a successful implementation of a change program. In line with recent theories on paradoxes and tensions in organizational change, our empirical outcomes imply that effective leadership of change requires the conscious acceptance of uncertainty and tensions between opposite options in key decision areas.

**Keywords:** organizational change; uncertainty; tension; grounded theory; change leader; automation

## 1. Introduction

The focus of this grounded theory study is organizational change in the context of implementing automatic parcel sorting systems in a global parcel delivery company, with the industrial engineering function (IE) acting as the change leader. We show that a key—but often neglected—aspect of successful change is the active detection and management of tensions stemming from uncertainty. By recognizing and managing uncertainty and tensions, the change leader plays an orchestrating role with respect to the differing views and expectations of the various corporate actors involved in the change process, gradually facilitating organizational learning and successful implementation of the change program.

Parcel delivery is a leading segment of the continuously expanding logistics industry, with a current annual growth rate of around 10% and global shipping volumes of almost 100 bn parcels per year [1]. Against this backdrop, the few global players in the market are undergoing rapid growth but also coming under strong competitive pressure to innovate with a view to expanding their capacity, transforming their business model, and enhancing the quality of their services.

Parcel sorting is a complex operational process performed in a dedicated facility called a hub. The hub is a physical place that receives stacks of incoming parcels from multiple points of origin. Parcels are unpacked, regrouped, and repacked by destination.

The present analysis is based on an extensive examination of organizational change processes across three European hubs of a global parcel delivery company where automatic sorting had recently been introduced. In the course of a three-year grounded theory study, which included three site visits and 43 interviews with internal informants at different levels of the organization, the authors gradually developed an in-depth understanding of a

complex corporate change program aimed at expanding capacity and radically transforming the parcel sorting process.

This grounded theory study, whose methodological framework we examine in detail in the next section (research methodology), yielded two different levels of outcomes: the first (Section 3: Research outcomes: the emergent substantive theory of change management as tension management) is a grounded conceptual mapping of the organizational change process, where the role of the industrial engineering function in leading the change process from an initial state (manual sort) to a final state (automated sort) is described in terms of the objectives, benefits, and drawbacks of automation, the main actors, and the most significant challenges faced. We offer a detailed analysis, grounded in the empirical data, for each of the proposed conceptual categories. The second level of outcome, a theoretical scale up from the first [2], is presented in the Discussion section and consists of an explicit analysis of the diverse views and narratives about uncertainty, the emergence of tensions, and the orchestrating role of the change leader. As one of the first field studies in this area to be grounded in in-depth empirical analysis, our work may offer a significant new contribution to the emergent debate on uncertainty and tensions in change management. Following a typical grounded theory approach [3], we postpone our theoretical background, theoretical extension, and contribution analysis until the Discussion section, where we relate our research to the ongoing academic debate and highlight the theoretical implications of our outcomes. In the concluding section of the paper, we discuss the managerial implications of our findings as well as potential future lines of inquiry.

## 2. Materials and Methods

This empirical investigation was carried out according to a Glaserian grounded theory (GT) methodological approach [2,4]; for an application of GT in the parcel delivery area, see e.g., [5]. Allowing empirically grounded theoretical concepts to emerge gradually was an especially fitting strategy given our open-ended, exploratory study. The Glaserian approach facilitates the emergence of empirically grounded theoretical concepts, in keeping with the researchers' initial exploratory perspective. With this approach, different views, perceptions, and interpretations can emerge, playing a key role in the final outcome. With respect to GT-informed research in general, the GT methods in this study were not used as tools for cross-case analysis, nor to complement other methodologies, but rather as a complete methodology for conducting in-depth interpretive analysis involving different actors and perspectives within a complex organizational setting.

At the outset of our study, the company in question had recently initiated a global automation process at its principal parcel sorting hubs, making it highly suitable for a multi-site, multinational investigation of the dynamics involved in organizational change. We focused our research efforts on three plants in two European countries, which varied in size and capacity but shared a common corporate identity and culture, as well as an urgent need to expand their overall capacity and range of services. The three sites were undergoing change at different scales of complexity and encountering different managerial challenges. Taken together, they therefore provided us with the data required to conduct an in-depth integrative analysis. The main focus of our inquiry was how the change process was being managed. Hence, we recruited the project managers—who in this company were appointed from the Industrial Engineering Department—as our key informants. Throughout the study, we constantly compared the PMs' points of view with those of the other corporate actors with a significant role in the change process: the hub directors and hub managers. During our on-site visits and interviews, we also elicited and recorded the perspectives of those in operational roles. We visited all three sites to observe their current functioning firsthand, as well as administer 22 face-to-face interviews and 21 conference-call interviews with three IE-project managers, three plant directors, three hub managers, and three supervisors, in addition to informal exchanges with operational staff during plant visits. We drew additional data from internal reports, presentations, and emails as well as corporate website pages and documents, supplementing these with material from

secondary sources including external reports on the company and press coverage. This is not an experimental study with the direct involvement of patients or volunteers, and no approval by an ethics committee is required. In any case, explicit verification and consent by the anonymous company involved were obtained at different stages of evolution of the original study.

An overview of the research process is provided in Figure 1 below. This was a multi-year and multi-site investigation: preliminary interviews were initiated in early 2016, followed by three major site visits and multiple cycles of data collection, data analysis, and theoretical framing over a three-year period. The final on-site interview took place towards the end of 2017, while the final data analysis, theoretical integration, and validity analysis took one additional year.

| | **Initial setup** | **Onsite visits** | | **Framing** |
|---|---|---|---|---|
| **data collection** | Preliminary interviews with the gatekeeper IE1<br><br>Telephone interview with IE2 | Visit to Plant Alpha. Interviews with IE2, two hub managers, three supervisors | Visit to Plant Beta. Key interview with Plant Director<br><br>Visit to Plant Gamma. Interviews with IE1, IE3, and hub manager | Follow-up interviews and further theoretical sampling with IE1 |
| **data analysis** | Shared interests; key elements of the automation program under analysis; key plants and activities to investigate; draft research plan; theoretical sense making; setup of interview guide | (Plant Alfa): first emerging key themes. Identified tensions; new interview guide | (Plant Beta): confirmed emerging themes; transcription, memoing and open coding<br><br>(Plant Gamma): coding of new material for data comparison and theoretical sampling | Selective coding; theoretical coding; theoretical integration; saturation and credibility analysis<br><br>Research output evolving towards a substantive grounded theory |

**Figure 1.** An overview of the research work.

A key aspect of grounded theory research is the so-called "constant comparison" of the empirical data collected and emerging concepts [3,4]. Data collection and conceptual analysis are not distinct and consecutive, but rather circular and interrelated. As a consequence, each of the three major research stages represented in Figure 1 (initial setup, onsite visits, and theoretical framing) required several iterations of the data collection (top) and data analysis (bottom) processes.

### 2.1. Initial Setup

We first established a relationship with a key industrial engineer (IE1: Industrial Engineer 1) at the company via a series of informal meetings. We next arranged three preliminary interviews: specifically, two face-to-face interviews with IE1 and a conference call with IE2 (the chief regional industrial engineer) that was also attended by IE1. A face-to-face follow-up session with IE1 helped us to clarify and confirm our notes. During all the interviews, discussions, and telephone calls, we both took separate notes in parallel. Immediately after collecting each new portion of data, we reviewed, discussed, and merged our notes with a view to identifying the main research themes grounded in our observations. To this end, we conducted "thematic coding", as treated in [2]. Specifically, we identified and noted down, then coded and linked, the themes emerging from the data (using the software application Atlas.ti [6] in order to build initial thematic concept networks, which we do not report here for the sake of brevity—see [7] for a detailed account. The main outcomes of the setup phase of the research are represented to the bottom left of Figure 1: developing a shared interest with the key informants, gathering information about the key elements of the automation program under investigation, identifying key plants and

activities to investigate, collating our first impressions for theory building, and developing an interview protocol and organizing the first visit.

### 2.2. Onsite Visits

We visited three plants, in two different countries. Plant Alfa is a medium-sized international hub, located in Southern Germany, with a capacity of about 30,000 parcels per hour and about 750 employees. Plant Beta is a major European international airport parcel sorting center, located in North-West Germany, where all air shipments to and from Europe are handled. Its sorting capacity is around 190,000 parcels per hour, and it has about 2100 employees. Plant Gamma is a relatively small terrestrial hub located in central France, with a capacity of about 10,000 parcels per hour and about 160 employees.

At Plant Alfa, we carried out two interviews. First, a two-hour interview with Industrial Engineer 2, followed by a one-hour interview with two hub directors (hub director 1 and hub director 2). We also had the unexpected opportunity to conduct two additional, unplanned informal interviews with three supervisors.

The data gathered during the visit were then validated and integrated in a follow-up interview with Industrial Engineer 1. An analysis phase followed: this involved developing a provisional preliminary framework with our proposed next research steps. We interviewed Industrial Engineer 1 again with a view to sharing this tentative framework as well as discussing the purpose and structure of a visit to a second plant. We chose Plant Beta and arranged to visit it. At Plant Beta, we conducted a two-hour interview with the Plant Director (PD1), and a follow-up interview with Industrial Engineer 1. Afterward, we analyzed the newly collected data, producing a more detailed set of research themes. Our visit to Plant Gamma, consisting of a two-hour live discussion with Industrial Engineer 1, followed by two one-hour interviews with the hub director and the project manager (Industrial Engineer 3), allowed us to engage in extended data collection and comparison, and to significantly advance our theoretical sampling.

### 2.3. Framing

Later on, we produced a full textual transcript of the recorded interview conducted at Plant Beta and attached descriptive labels to all individual textual tags in the text using Atlas.ti [6]. The descriptive labels were produced by one of the authors and then discussed and modified with the other author. This step is often referred to as open coding in grounded theory methodology.

In order to identify meaningful associations among the codes, the coding labels were copied onto post-its of various colors and sizes and grouped together into meaningful categories that gradually came to form a three-level conceptual hierarchy (selective coding).

We next set out to identify meaningful relationships linking the categories to one another (theoretical coding). This evolving system of concepts, categories, and relationships gradually took shape on a series of posters covered with post-its of different colors representing different concepts, providing a basis for theoretical discussion, further data collection, and theoretical sampling. We later transferred the resulting set of grouped and linked paper-based concepts back into the original digital format. After several iterations, our investigation and analysis of three major industrial automation projects in Plants Alfa, Beta, and Gamma, respectively, converged towards data saturation, i.e., "the point in coding when you find that no new codes occur in the data [2] (p. 194), yielding the substantive theory summarized in Figure 2 below.

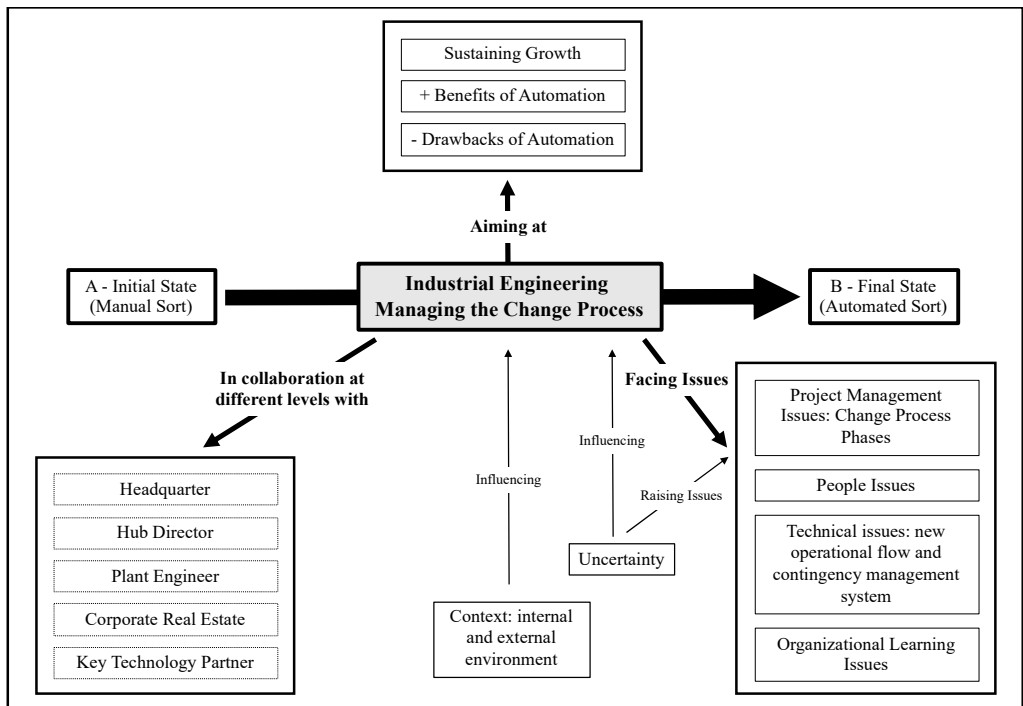

**Figure 2.** A conceptual map representing the emergent substantive theory of change management as tension management.

## 3. Research Outcomes: The Emergent Substantive Theory of Change Management as Tension Management

The first outcome of our analysis was a conceptual map of the organizational change process, as represented in Figure 2.

Typically, there are two key steps in the parcel sorting process (Figure 3 below). The first is identifying incoming parcels (represented in the bottom part of the picture) by origin and destination, via automatic identification portals. The second is assigning and moving parcels to their destination groups, as represented in the upper part of Figure 3.

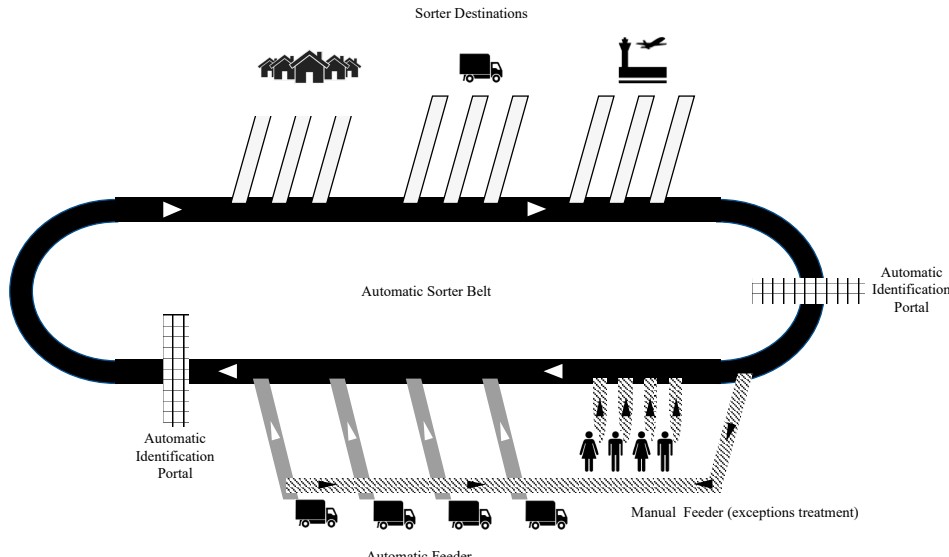

**Figure 3.** The automated parcel sorting process.

Before automation, parcel handling and sorting were fully manual procedures: parcels were moved around the hub on conveyor belts (the only mechanized step in the process), but parcel loading, sorting, and unloading were performed manually by human operators.

Following the introduction of automation (Final State B), this operation requires little or no human intervention. A complex system of scanners, photo-eyes, and cameras identifies parcels and reads their destination data so that they can be automatically transferred to the appropriate group. Migration from State A to State B was accomplished via a change program aimed at enhancing organizational growth by unlocking multiple benefits of automation while minimizing its drawbacks (see Figure 2, top). This program was led by the industrial engineering function, in collaboration with a range of key actors across different levels within the organization (Figure 2, bottom left). The conceptual map shows that change was characterized by uncertainty, raising issues that were managed by IE but also shaped by contextual factors (Figure 2, bottom right).

*3.1. The IE's Role in the Change Process*

Our analysis suggests that the IE, in acting as a leader of change, evolves towards the role of business partner, going beyond his/her traditional remit as technical staff (Figure 4).

| FROM the traditional IE Core Tasks | TO the new IE Core Tasks |
|---|---|
| Operating Plans | **Business Growth Support**<br>Service analysis, business planning, service enhancement, customer orientation, sustainability. <u>New competencies required: business case analysis, market and trends knowledge, financial basics</u> |
| Implementation | Operating Plans<br>Design and mastering of operating plans, staffing plans, coordination of internal and external operations. <u>New competencies required: negotiation, "out-of-the-box" thinking</u> |
| Work Measurement | **Operational Improvement & Implementation**<br>Improved specification of operational processes, cost savings, gains in efficiency, reduced environmental impacts. <u>New competencies: leveraging dissatisfaction as an opportunity for learning and improvement</u> |
| Training | Work Measurement & Training<br>Evaluation of processes and procedures, error detection and analysis, operations training. <u>New competencies: persuasion, accountability, evaluation of opportunities and constraints</u> |
| | **Systems and Operational Technology**<br>Data management, systems implementation, execution of technology projects. <u>New competencies: innovation-driven performance enhancement, self-training, IT awareness</u> |

**Figure 4.** The evolving organizational role of the industrial engineering function.

Two key categories emerged from the GT analysis (Figure 5): IE social role and IE technical role.

Traditionally, the IE's technical role comprises three main components: standardizing processes, developing plans for the management of both operational and contingency situations, and setting up flows.

The social role of an IE on the other hand entails managing behaviors on the ground. Thus, the IE's plan for implementing change needs to include strategies for addressing resistance and behavioral inertia (code: role of IE handling resistance to change), such as the tendency to maintain direct communications instead of making use of telemetric instruments.

Managing behaviors on the ground also requires the presence locally of an IE member of staff (code: role of the Regional Industrial Engineering Department) whose main task is to detect unforeseen events, manage inertia and resistance, and collect the information required to monitor progress. During the stabilization phase, the local IE is required to supervise the application of performance management approaches and tools and define a set of KPIs.

Indeed, the social dimension of change generates uncertainty—which is additional to technical uncertainties—at the initial planning stage. To address such uncertainties, as discussed below, the social role of the IE also encompasses stimulating and enabling

experimentation, tinkering and tuning, and a strong element of negotiation (code: role of IE pushing toward emergent).

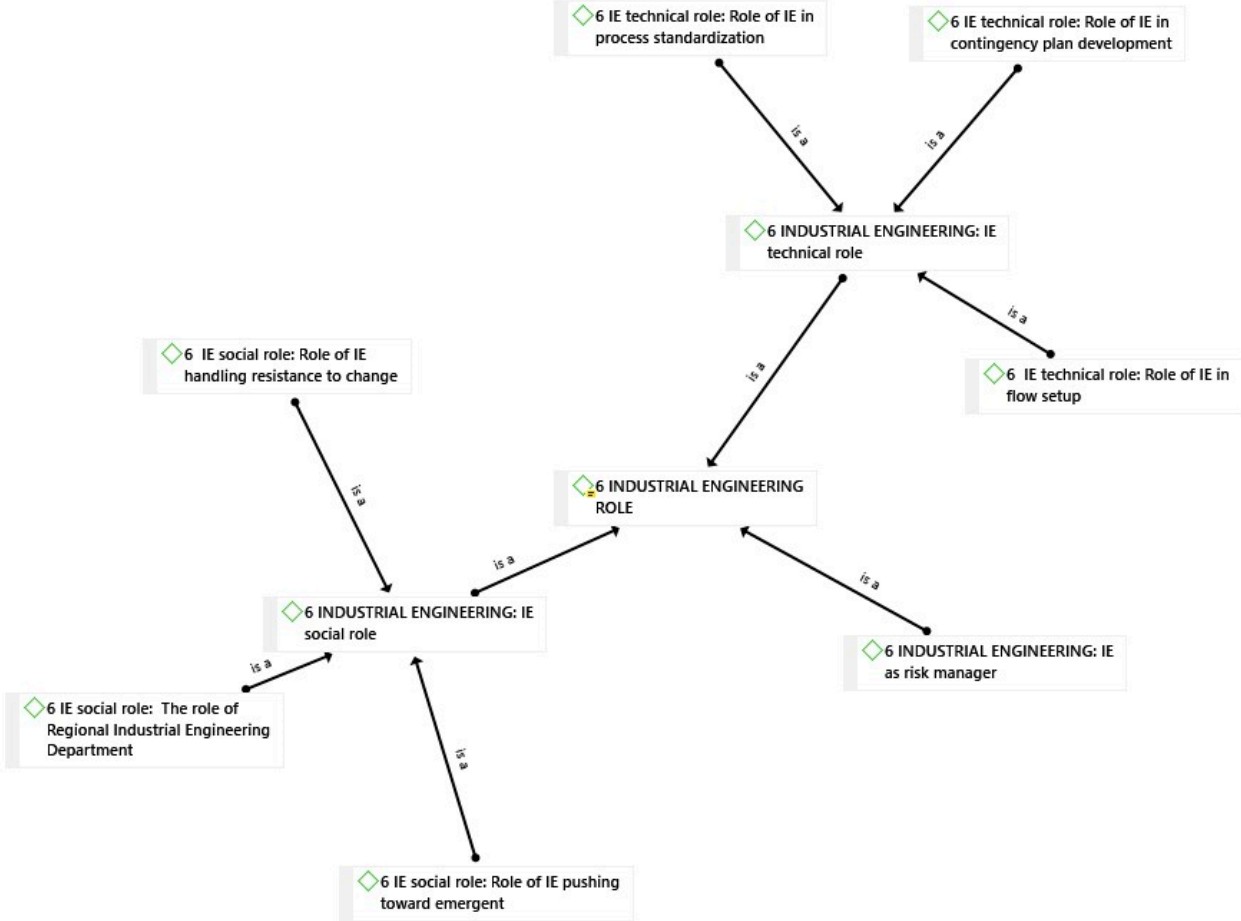

**Figure 5.** Conceptual map of the role of the industrial engineering function.

### 3.2. The Relationship between the IE Function and the Other Internal and External Organizational Actors Involved in the Change

Those involved in the execution and overseeing of operations expect the IE to provide a complete implementation plan at the initial planning stage, and this pushes the IE to make explicit provision for dealing flexibly with uncertainty, which must be negotiated with the hub manager and other internal and external actors involved in the change, as depicted to the left of Figure 2: headquarters, real estate, plant engineer, and key technology partners. The IE's central relational position demands skill in understanding and mediating among the diverse perceptions, perspectives, and priorities expressed by the above-mentioned actors at different levels. This orchestration role also requires the physical presence of an IE resource—such as a Regional IE, as mentioned above—on-site at all times.

### 3.3. Guiding Principles: "Sustaining Growth", "Benefits of Automation", and "Drawbacks of Automation"

From the interview data, a set of guiding principles (see the top half of the conceptual map in Figure 2) emerged as motivating and shaping the change process. More specifically, we identified two major categories of strategic driver: "growth" and "benefits and drawbacks of automation".

### 3.3.1. Sustaining Growth

A fundamental driver for the organization in our study was the goal of attaining significant and sustained expansion of service volumes ("growth"). Technology was being leveraged to sustain volume growth while simultaneously enhancing the content and quality of the company's services and improving productivity. Throughout this complex organizational expansion and change process, technology-enabled benefits were to be pursued while the associated drawbacks were to be carefully addressed.

### 3.3.2. Benefits of Automation

Two major subcategories of benefits emerged from the analysis: value generation and cost saving.

Value generation. Technology-enabled change can facilitate, on the one hand, the development of new high-value services, particularly for the growing market segment of e-commerce; on the other hand, it will allow the company to improve the quality of its existing services. Specifically, new value will be generated via the capacity to handle greater organizational complexity, substantial gains in the speed of the logistic flow, enhanced capability to deal with external contingencies (e.g., unforeseen events such as weather changes that generate spikes in demand over the short term), and increased potential capability to further invest in new technologies such as AI-enhanced customer interaction channels or IoT and smart devices.

Cost saving. The benefits grouped under the "cost saving" category were more in line with those of traditional automation projects: the radical increase in capacity and flow speed had produced evident economies of scale, with accompanying gains in productivity. At the operational level, the "reduction in the number of times packages are touched" was having a marked positive impact on work productivity, while reaction times to frequently occurring contingencies such as package jams had greatly improved.

### 3.3.3. Drawbacks of Automation

Two main drawbacks emerged in relation to the change program: "reliability and associated risks" and "data quality".

Reliability and associated risks. Typically, technological innovations are associated with poor reliability during the initial phases of their adoption. We found evidence of this phenomenon in the change project under study. Issues with reliability can limit or delay the technological transition. The level of operational risk must in any case be maintained below a level that is sustainable for the organization.

Data quality. A key challenge for this change program was the issue of data quality. For the automation system to work, well-defined technical specifications must be met, for example, to ensure that barcode labels are uniform in format and content. Such specifications must be complied with both internally and externally to the organization, in other words, throughout the entire service supply chain. Achieving this requires complex inter-organizational coordination that is often difficult to negotiate.

### 3.4. People Issues: Generating a New Mindset

Among the various issues faced by the change leader, we might expect technical issues to dominate, given that the change being studied is technology-driven, and given that IEs have traditionally fulfilled a mainly technical role. Yet our analysis suggests that people issues, depicted in Figure 6 below, are among the most critical and complex, implying a significantly different role for the IE compared to the past, as discussed above.

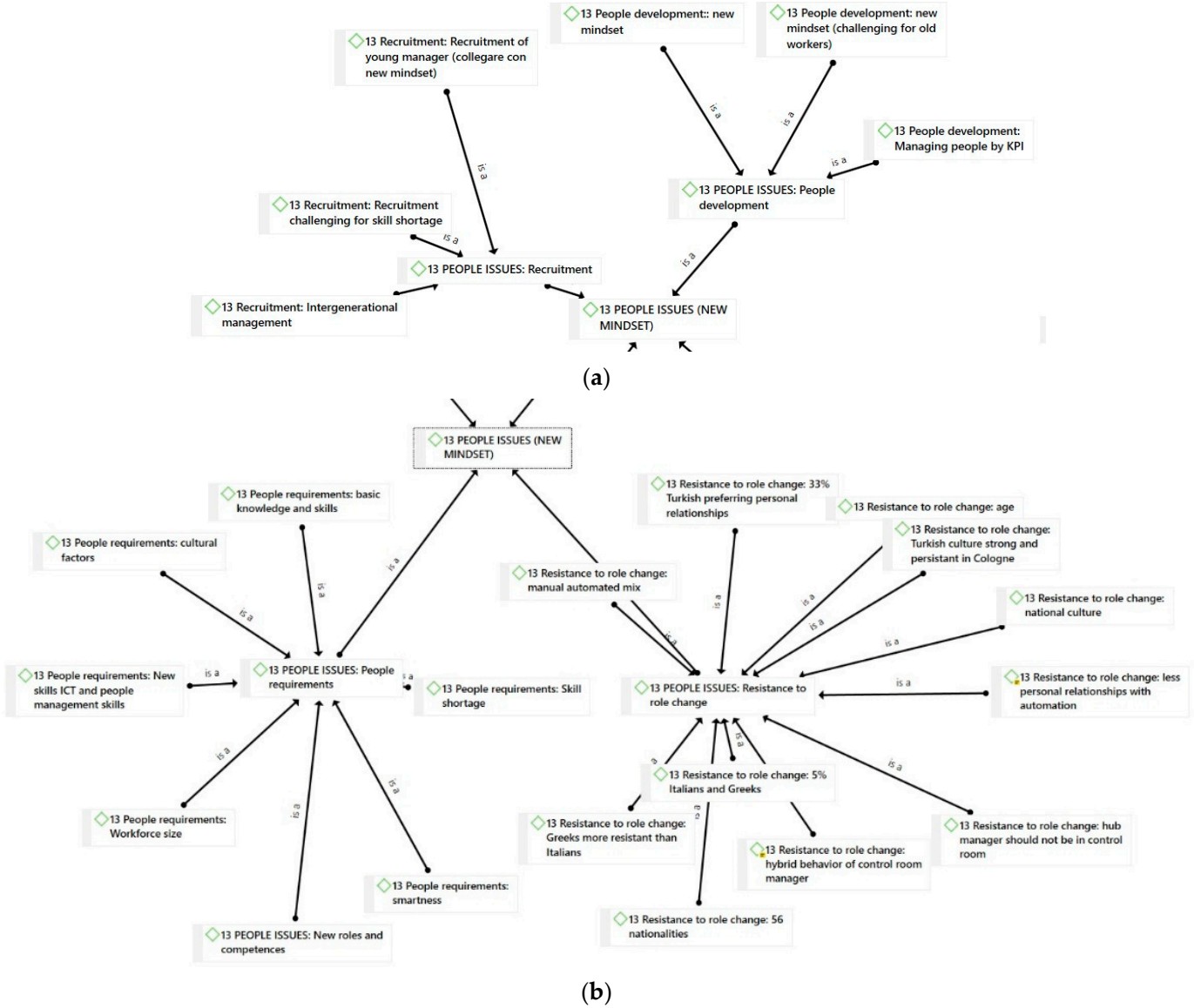

**Figure 6.** (**a**) Conceptual map of people issues (top side); (**b**) conceptual map of people issues (bottom side).

Our data suggest that promoting a new mindset is the core "people issue". The new system requires staff not only to perform new activities, but also to embrace a radically different mentality, with a shift in focus from routine operational execution towards information-driven sense-making and problem-solving, all within a broader and complex information process. A comprehensive understanding of the interdependencies among organizational processes is required, together with technical and digital awareness. Shifts in mentality are rarely immediate: operatives should be given time to embrace and adjust to the new mindset before being actively involved in the development of new solutions and encouraged to play an active role in the new system. Leading and fostering such a process requires the change leader to leverage multiple dimensions of people management, as illustrated by the conceptual map in Figure 6 below. Such dimensions are represented by the four main categories on the map: new people requirements, recruitment, people development, and resistance to role change.

Besides overseeing a shift in mentality, the change leader needs to ensure uniformity in the new mindset at the global level, while taking local specificities (cultural, competence-related, path-dependencies) into due account. This generates a tension that we might label "corporate vs. site", between standard corporate directives and specific local needs, as further described below.

### 3.5. Project Management Issues: The Phases in the Change Process

Textual analysis of our interviews with the IE with responsibility for the project suggested that there are three key phases, represented in Figure 7 as main conceptual categories: analysis, planning, and implementation. Each of these categories is further defined by specific codes that feature on the conceptual map.

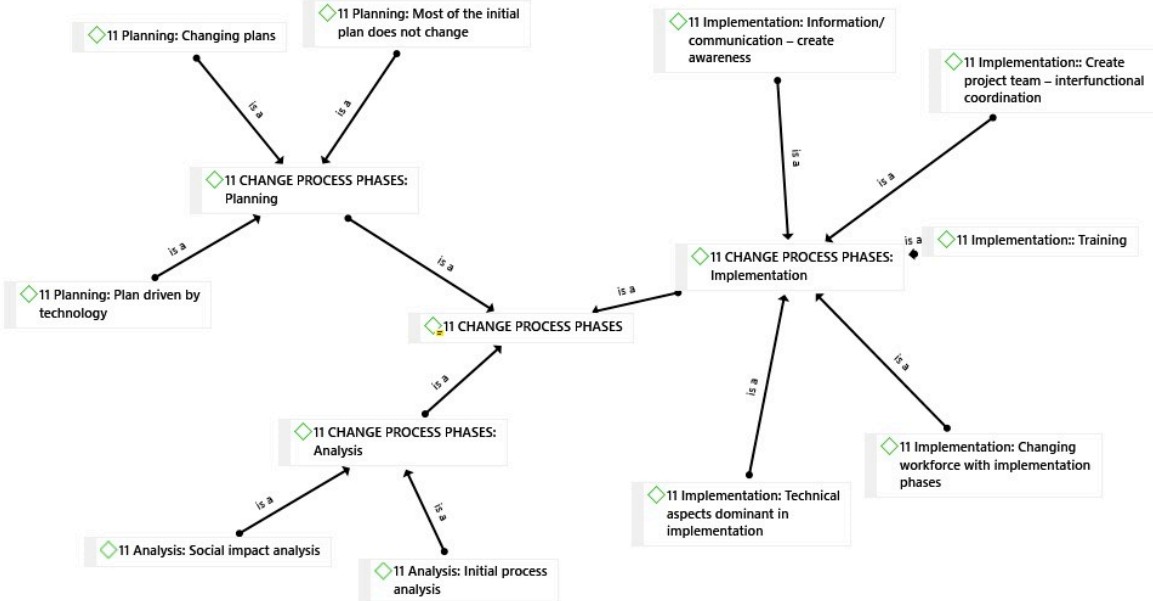

**Figure 7.** Change process phases.

The three dimensions of analysis, planning, and implementation form a logical sequence. However, the change leader's challenge is to manage all three dimensions while taking into account emergent factors generated by unpredictable social interactions.

### 3.6. Technical Issues: New Operational Flow and Contingency Management System

The so-called "technical issues" identified at the bottom right of Figure 2 are those most comprehensively supported by the empirical data. They concern the technical implementation and management of the new automated sorting system, which is no longer based on the visual monitoring of flows but rather on the collection and analysis of flow data that is processed in the control room. We analyze and discuss the technical aspects of this transition under three headings: the management of flows and contingencies prior to automation (State A), the new system post-automation (State B), and issues related to the implementation and management of the new system.

#### 3.6.1. Operational Flow and Contingency Management before Automation (State A)

In the respondents' terms, an operational exception is any unexpected situation arising in the hub (as a function of variations in package volumes, weather conditions, destinations, delays, failures, etc.). An exceptional occurrence with the potential to disrupt the normal flow of operations if not properly handled is termed a contingency. Contingency management entails devising processes to fit any contingency.

In State A, (Figure 8 below) contingency management is "ad hoc" (category: ad hoc contingency management (physical and visual). For example, in the case of belt failure, the supervisor immediately sets up an emergency workaround, using alternative belts to ensure the continuity of the sorting process. He/she is a middle manager (code: visual supervisors as middle managers) who, through visual inspection, continuously interacts with sorters, using radio communication to collect information and issue instructions for addressing failures (code: totally manual voice coordination). Human control dominates (category:

relevance of human factor): relational and behavioral skills are more important than technical skills (code: skill change less technical more behavioral). Decisions are informed by visual inspection of the sorting line, which is crucial to detecting and understanding failures, evaluating alternatives, and identifying appropriate real-time solutions. The decision-making process is creative, informal, flexible, and embedded in experience (code: creative flexibility but less efficient). Contingency plans exist but are not embedded in a software procedure.

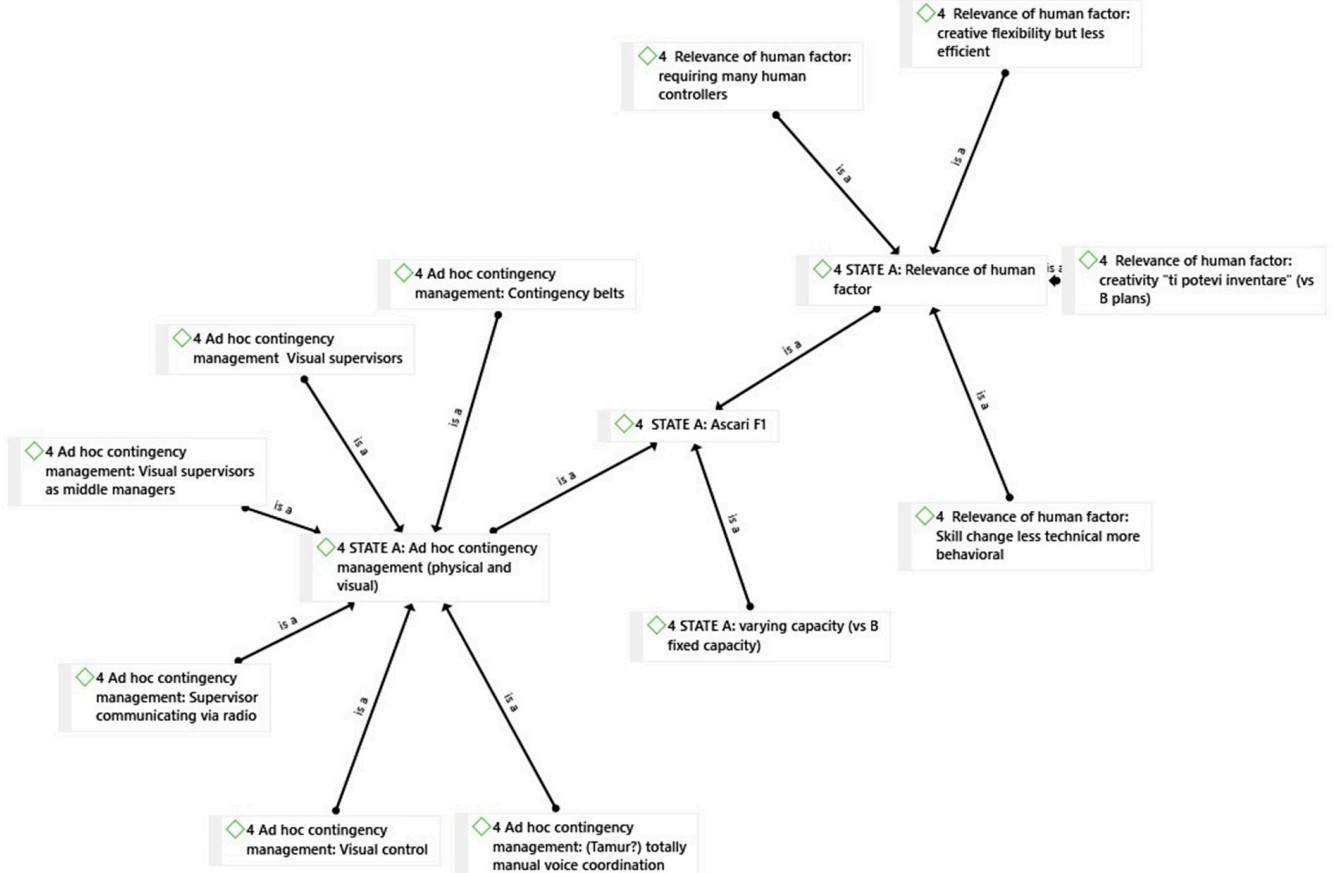

**Figure 8.** State A.

### 3.6.2. Operational Flow and Contingency Management after Automation (State B)

In State B, contingency management is "telemetrically-based" (category: telemetric based contingency management). In the belt failure scenario, a telemetric system automatically generates alerts and detailed technical data about the problem (code: telemetric control). There is a physical place, the control room, where advanced software applications for the remote management and configuration of the sorting lines are run. Management of a contingency proceeds through three stages: alert generation, alert analysis, and selection of the most appropriate contingency plan (code: solution from the control room). Ex-post data analysis informs continuous improvement in predictive maintenance planning and performance, as well as adjustments to contingency plans. The introduction of a control room has implications for people management, in terms of human resource requirements (category: human factor in the control room), as expressed via the codes on the right-hand side of Figure 9.

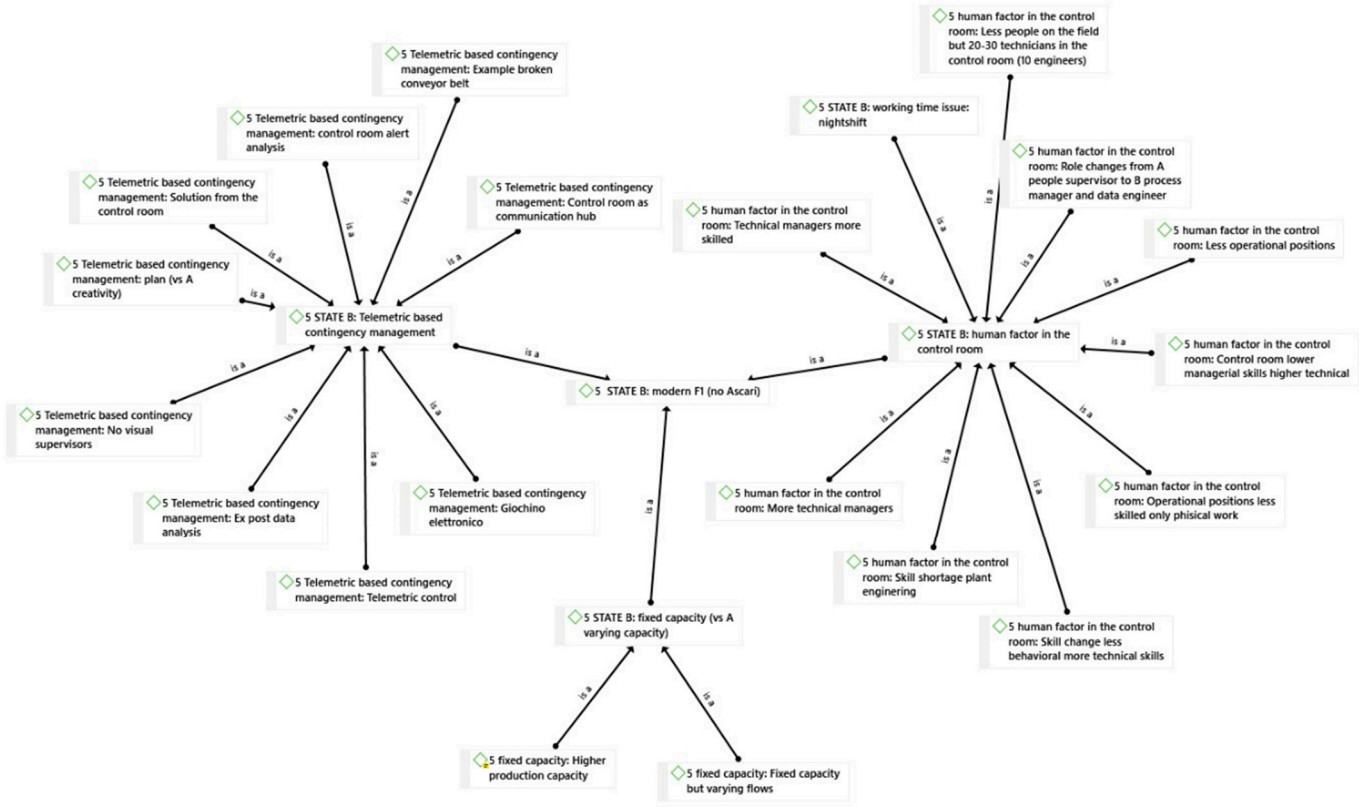

**Figure 9.** State B.

### 3.6.3. Implementing and Using the New Flow Setup and Contingency Management System

The new operational model introduces the need to design and implement a new social and technical system for managing flows and contingencies. From a social point of view, such a system places significantly different demands on the organization's human resources, as discussed in the people issues section. From a technical point of view, two key innovations come into play, shown in Figure 10 as the main analytical categories: system-based contingency diagnosis and predefined contingency plans.

System-based contingency diagnosis: Operations are monitored in real time via a system of sensors that automatically and continuously collect data (code: contingency diagnosis based on sensor data). When an irregular situation occurs or critical thresholds are exceeded, a diagnostic intervention is required and will be based on analysis of the collected data. Diagnostic testing will be carried out by the control room engineers (code: contingency diagnosis in the control room) using a sophisticated dashboard system that makes the data intelligible by producing predefined graphic and numeric indicators. Within this system, the role of the control room engineers is clearly defined insofar as it is limited to operating the diagnostic system (code: ownerships clearly defined for IE and operations). At the same time, design or modification of the diagnostic system falls under the responsibility of the industrial engineering function, but can benefit from the cooperative input of the operatives on the ground, especially during the problem setting phase. Thus, diagnosing contingencies no longer depends on the skill, past experience, and individual characteristics of the operatives on the ground (as was the case in State A) but is embedded in the social and technical system itself (code: dependent on system embedded knowledge). This is because the knowledge used to make the diagnosis is now based on organizational interpretive capacity and indicators of critical states that are incorporated into system procedures.

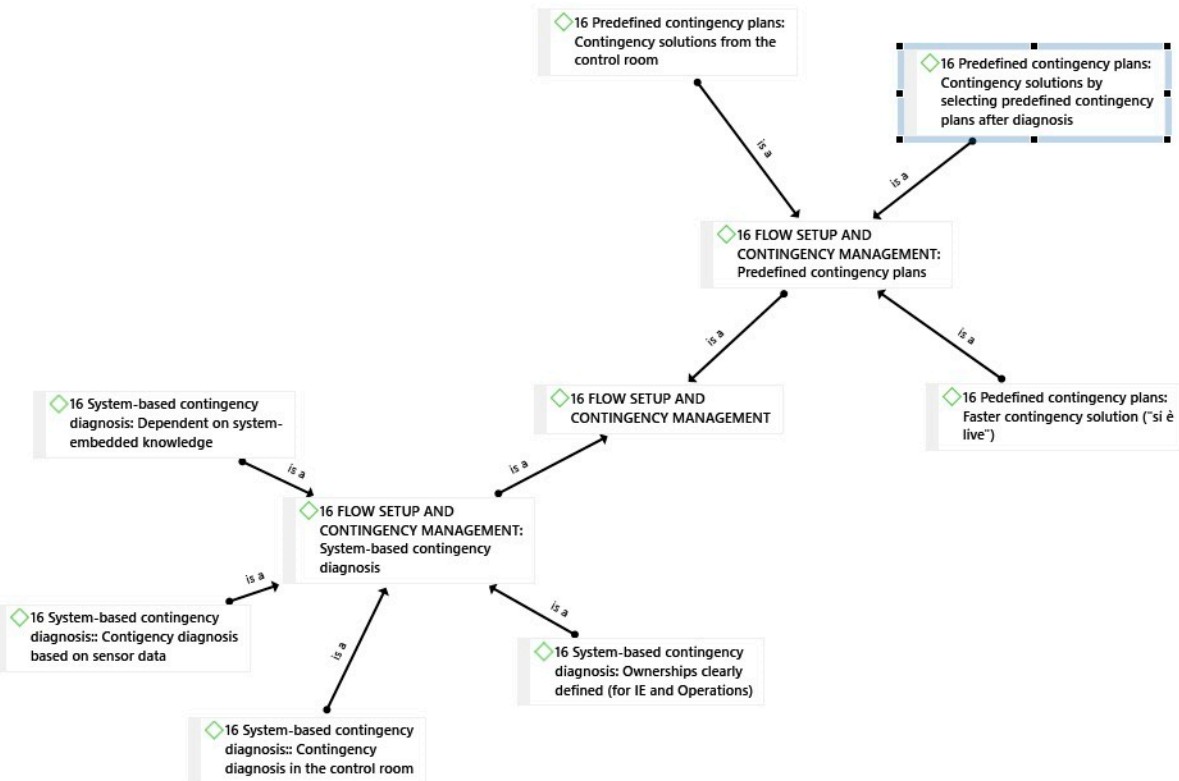

**Figure 10.** New socio-technical system issues: flow setup and contingency management.

Predefined contingency plans. Managing the irregularities detected by the control room entails making a real-time decision (code: faster contingency solutions), in terms of matching the present scenario with one of a suite of predefined contingency plans specifying the optimal responses to different sets of circumstances (code: contingency solutions by selecting predefined contingency plans after diagnosis). If there is no existing or fully appropriate contingency plan for the current issues, the local IE will be required to intervene ad hoc, in collaboration with the sorting operatives, to develop a new contingency plan or modify a present one, as already discussed in the section entitled "The IE role in the change process". Compared to the situation prior to automation, the resolution of irregular or unusual situations should be quicker and more reliable, in that it is based on a system of embedded knowledge.

In sum, under the new socio-technical system, almost all decisions are now made in the control room, which has become the "brain" of the sorting process (code: contingency solutions from the control room), while in State A, decisions could be made at any location, and informal solutions that had not necessarily been coordinated at the systemic level could be adopted.

### 3.7. Context

A set of positive contextual factors contributed to the success of the change process, as borne out by the main categories shown in Figure 11: External environment and Internal environment. The characteristics of the internal context correspond to the analytical codes on the right-hand side of the figure: a pattern of strong work involvement, empowerment, and pride, rooted in a participatory corporate culture, leading to high acceptance of change.

Factors related to the external environment included the fact that the change was introduced during a period of market expansion (code: role of market growth), a suitable level of readiness on the part of all the actors involved in the broader inter-organizational process from initial shipping to final delivery of the parcel (including e-commerce shoppers, intermediaries, producers, carriers, etc.) (code: market maturity for automation), and the degree of acceptance of automation attained by the company's business customers,

including the kind of organizational mindset required to ensure the efficient and effective implementation of automated processes (code: cultural aspects and automation acceptance by customers). A further success factor was the match between the level of automation required by the parcel service and that already implemented by many of its business customers (code: automation level matching with customer automation level).

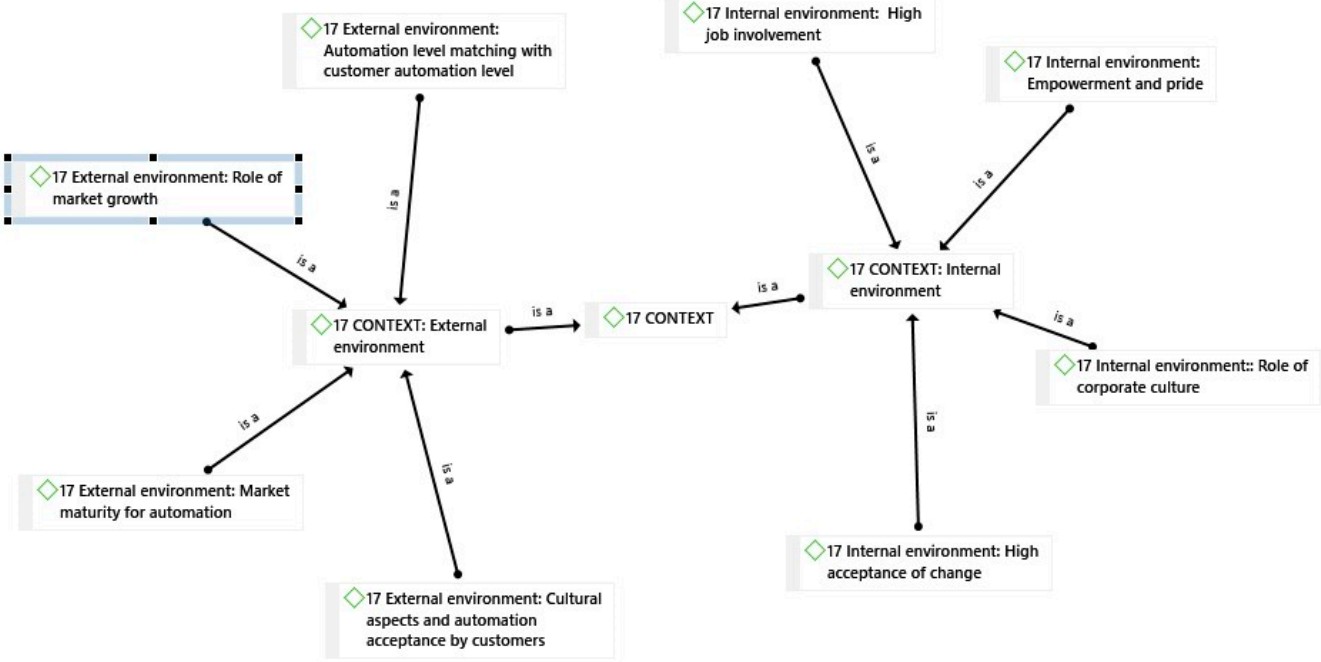

**Figure 11.** Conceptual map of contextual factors.

### 3.8. Uncertainty

The change process starts from manual State A with the aim of leading the organization towards automated State B, which is still partially unknown. The main sources of uncertainty include the human, relational, and social implications of developing new technical solutions for managing operational flows and contingencies, as observed in the sections on people and technical issues. This uncertainty leads to the coexistence of manual and automated solutions, with the attendant need to define where in the operational flow to introduce full automation and where to maintain manual or hybrid solutions. It is also important to decide what level of automation is ideally being aimed for, given that the most sophisticated level theoretically possible is not always the optimum solution.

Hence, there is a need to strike a dynamic balance between two opposite poles (totally automated, totally manual), which we term the "manual vs. automated" tension (see Section 4: Discussion, theoretical integration, and theory extension). These sources of uncertainty, represented in Figure 12, are caused on the one hand by initially unforeseeable situations and on the other hand by changes to the details of earlier plans. They are also present at the prior change planning phase (see the category "5 uncertainty: at change planning" to the bottom left of the figure), as well as at the subsequent change implementation phase (category "uncertainty: at change implementation" to the top right of the figure), as emerged from the coded empirical data.

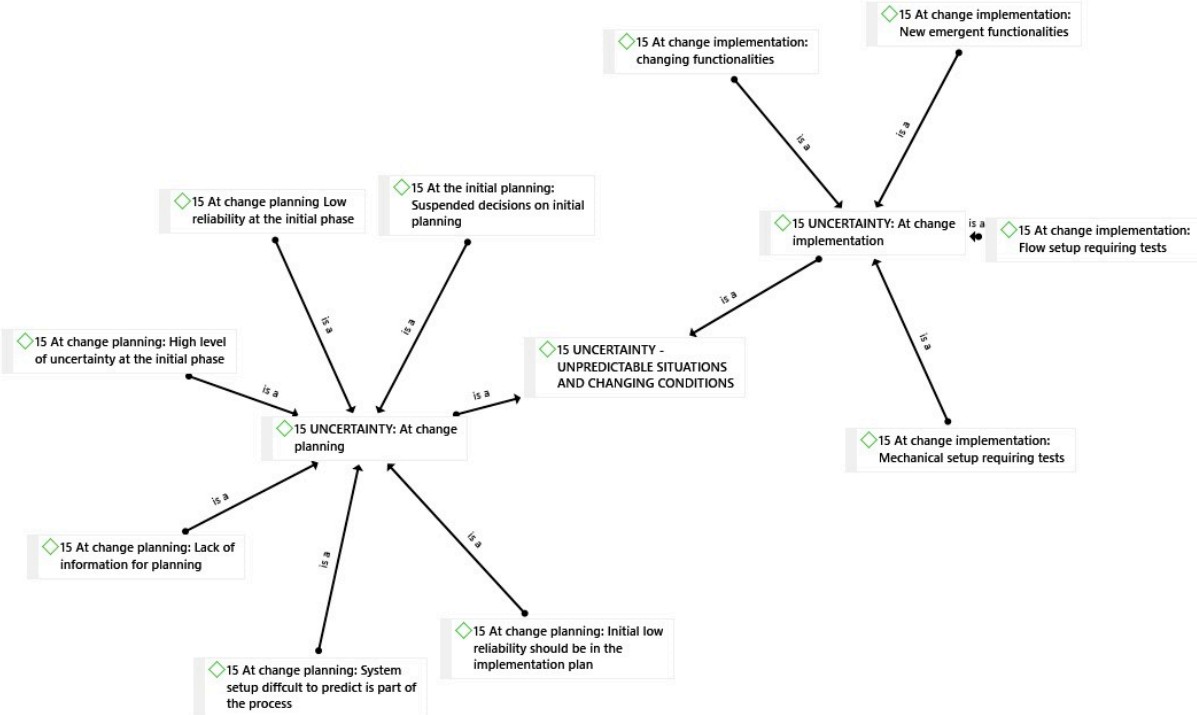

**Figure 12.** Uncertainty issues in the change process.

### 3.8.1. Uncertainty at Change Planning

The change planning stage is marked by a higher level of uncertainty associated with a lower level of operational reliability, given that a solidly structured implementation plan does not preclude the possibility of later defining details and confirming choices that have been deliberately left open and require experimentation on the ground before finalizing. Our respondents suggested that uncertainty arises because (1) the setup of a new system inevitably entails experimentation and fine-tuning, while (2) devising a perfect plan would require information that is not yet available. This scenario impacts the planning process: certain decisions are initially left on hold, while initial poor reliability must be explicitly taken into account when defining goals.

### 3.8.2. Uncertainty at Change Implementation

Respondents reported that uncertainty during implementation concerns both setting up the new operational system and assimilating changes or new functionalities discovered along the way. Throughout implementation, tensions associated with uncertainty tend to emerge and should be managed by the change leader. How he/she does so has implications for the outcomes of the change process. These tensions, which we have labeled planned vs. emergent and corporate vs. site, will be outlined in greater detail in the next section (organizational learning) and analyzed in light of the existing literature in the discussion section.

### 3.9. Organizational Learning

The issue of organizational learning is closely related to the uncertainty, both social and technical, that characterizes the initial stages of change implementation. The learning process is illustrated on the left-hand side of Figure 13 below (category: reducing uncertainty by learning contingency management) and is brought into effect via both interactions within the organization itself (category: corporate knowledge sharing) and collaborative exchanges with outside partners (category: external environment knowledge sharing).

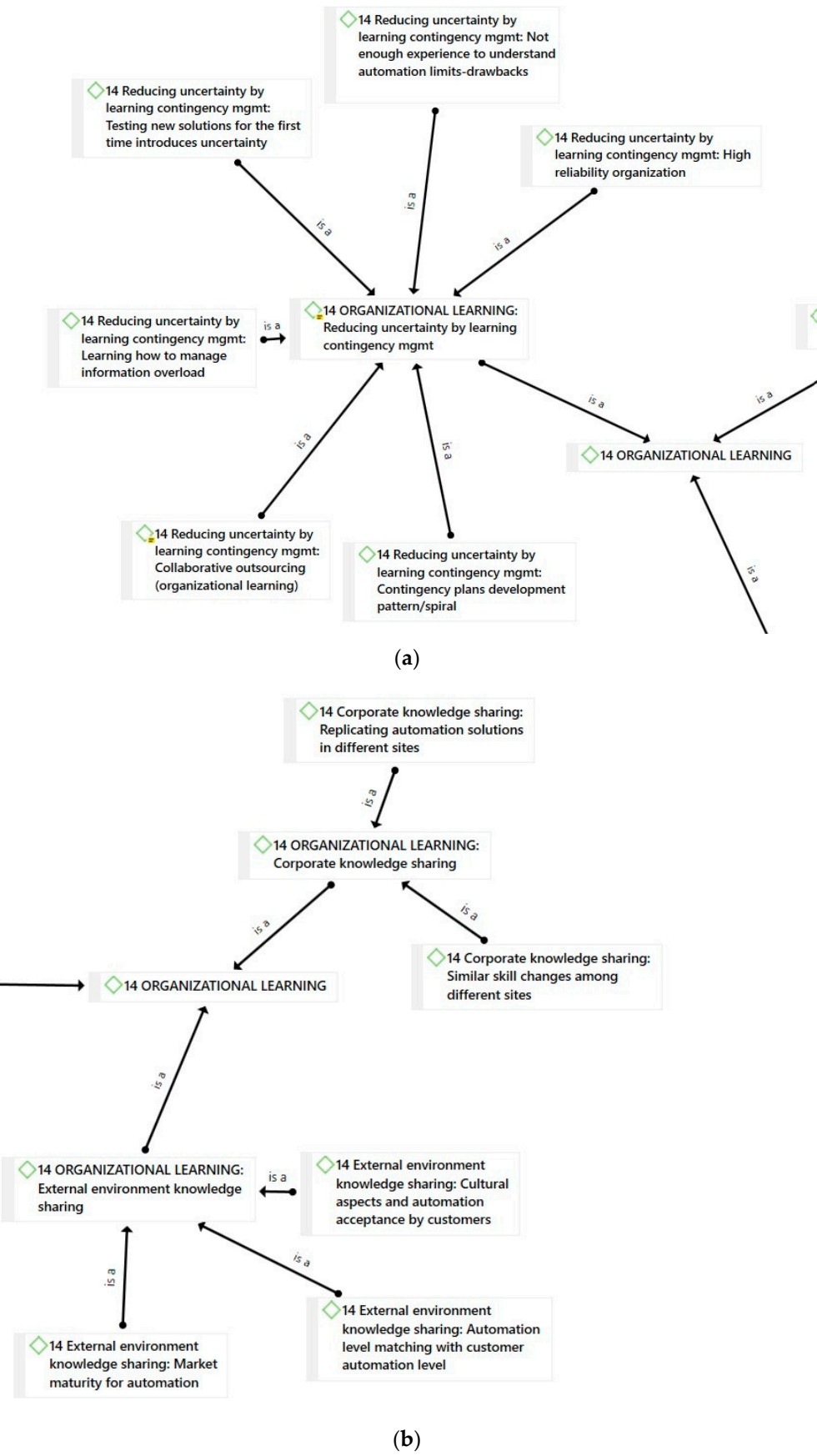

**Figure 13.** (**a**) Organizational learning (left side); (**b**) organizational learning (right side).

### 3.9.1. Reducing Uncertainty by Learning Contingency Management

Given that, especially in the initial stages, the organization has no consolidated body of experience informing it on how far it should push the automation process, experimentation is needed to develop an in-depth understanding of where and to what extent it will be more effective to introduce automatic solutions. This gives rise to a learning process (code: not enough experience to understand automation limits). A second aspect, which is often related to the first, concerns the planning of the change. The intentional inclusion in plans of areas of experimentation means that implementation will involve testing new solutions (code: testing new solutions for the first time). Again, we identified the need to strike a dynamic balance between two opposite poles (fully specified plan, no plan), which we term the "planned vs. emergent" tension.

In general, learning—from an initial situation characterized by great uncertainty and low reliability—progressively reduces uncertainty, contributing to the development of a highly reliable organizational model. This is achieved via cycles of experimentation that gradually diminish the critical and/or intentionally undefined areas of initial contingency plans (code: contingency plans development pattern/spiral), in part thanks to cooperation with partner organizations that, at some sites, play a key role in the learning process (code: collaborative outsourcing). In addition to defining contingency plans, another area of learning concerns the most effective ways to manage the risk of being overwhelmed by the flow of telemetric data generated by automated systems (code: learning how to manage information overload).

### 3.9.2. Corporate Knowledge Sharing

The new learning generates codifiable and transferable knowledge, allowing successful technical solutions to be implemented at multiple sites, while similar patterns of change in skill sets are also observed across different hubs. Once again, we observed a dynamic process of seeking a balance between two opposite poles, in this case between the need to find standard operational solutions that could be shared across the organization's different sites (corporate pole) and the need to take into account the peculiar features of each individual site (site pole). We termed this tension "corporate vs. site".

### 3.9.3. External Environment Knowledge Sharing

Furthermore, learning is acquired with the help of third parties, who are external to the organization but part of the value chain. Indeed, when change is introduced, it is critical to take into account the degree of maturity of the organization's markets and customers. In striking this balance, organizational culture and customers' propensity to accept new automated solutions also come into play.

## 4. Discussion, Theoretical Integration, and Theory Extension

The change process outlined in the previous section and represented in Figure 2 was identified via a grounded theory research methodology that enabled us to attain two objectives: first, the firm anchoring of our proposed concepts and categories in the data (interview transcripts and documents) that we collected in the course of the study, and second (and perhaps even more crucially), the emergence of plural voices and perspectives representing the different organizational actors involved in the change process. This plurality of voices and perspectives is of key importance when analyzing the uncertainty associated with the change process and how it was perceived and narrated by our informants. Specifically, the organization's dominant narrative tended to exclude the possibility of uncertainty. For example, at one of the observed sites, the hub director explicitly stated that the ideal scenario from his viewpoint would be to have a "perfect" implementation plan (which would be the responsibility of the IE function), while going forward the optimum final outcome would be a 100% automated system. This perspective was shared by the IE, in keeping both with a traditional IE approach and with the system of mutual expectations associated with the IE function. However, our analysis shows that, as discussed in the

section on uncertainty, implementing the change plan on the ground necessarily entails taking into account the human, relational, and social implications of developing new technical solutions [8–24]. People's reactions to a situation that is new to them and that cannot be fully represented or simulated in advance can in some respects only be known and managed as the new system is being experimented with and experienced [25–38]. This is particularly true, as discussed in the section on people issues, when new roles with high levels of autonomy and competence—such as that of control room operative—need to be introduced. On the other hand, as noted above, it is also necessary to establish what level of automation is ultimately to be pursued, given that the most advanced level theoretically possible is not always the optimum choice [39,40]. Furthermore, technologies are continuously evolving, and their application in the field often requires ongoing experimentation [41,42]. In sum, while the main narrative calls for the elimination of uncertainty by means of rational planning, in practice, uncertainty at both the human and technical levels is an inescapable aspect of the change process; it calls for hands-on experimentation whose outcomes will progressively inform decisions on how exactly to implement the automation system. In other words, it is crucially important to consider both technical and social aspects in organizational change implementation, as socio-technical studies have been emphasizing for a long time [43]. However, in contemporary organizations, social and technical aspects alone may not suffice to account for the complex emergent dynamics of organizational change, which are often influenced by contrasting perceptions of uncertainty. Effective socio-technical change should explicitly address the dynamics of uncertainty by substantially rethinking rational design practices or even removing the formal distinction between system design and implementation. "We use the term construction rather than design and implementation because approaches such as agile development and configuration of ERP systems do not distinguish between these activities. [...] Getting the construction right is not simply a matter of writing better system requirements. In the same way that requirements for a user interface cannot adequately express the richness of the interaction with a particular system, social and organizational complexity cannot be simply distilled into 'social' or 'cooperation' requirements" [44] (p. 13). Thus, in contrast to the organization's dominant narrative, the change leader deliberately leaves specific areas unplanned (i.e., certain decisions are put on hold) or earmarks them for review. By this means, the ineliminable uncertainties associated with the implementation of the plan can gradually be addressed, managed, and overcome.

Our analysis showed that managing uncertainty is particularly necessary in relation to three sets of competing poles, corresponding to the main types of decision to be put on hold or reviewed: manual vs. automatic, planned vs. emergent, and corporate vs. site. The change leader, while taking into account both the initial plan and needs emerging on the ground, gradually defines positioning with respect to each of these sets of poles, as experimentation on the ground produces evidence and generates both individual and group experience. The iterative experimentation process is thus centered around a set of tensions that are key to managing uncertainty and that allow the change leader, while apparently conforming to the corporate narrative of the "perfect plan", to respect the cornerstones of the initial plan yet take full account of and thoroughly address the areas of uncertainty. Tensions may often be related to the well-known phenomenon of resistance to change, but the nature and the sources of tension are well beyond resistance and are related to the wider areas of uncertainty characterizing organizational change.

Narratives and tensions are related: the coexistence of multiple competing narratives about the uncertainty surrounding implementation (hub director: it does not exist; IE: it exists) is associated with different expectations and negotiating positions with respect to the poles of manual vs. automatic, planned vs. emergent, and corporate vs. site. Managing these tensions, and thereby gradually defining a position with respect to each set of competing poles, is thus a central focus for the change leader. Ultimately, the change leader is in charge of a negotiation process, the outcome of which will be the progressive

shaping of a shared perspective among all the actors in the interest of ensuring effective implementation of the plan for change.

All in all, by managing the set of tensions just described, the change leader gradually overcomes uncertainty and brokers a degree of convergence among the perspectives of the various actors. The analysis of tension management proposed in this study shows significant affinities with existing perspectives in organizational studies. In a recent paper on the topic, Hargrave and Van de Ven (2017) [45] summarized developments in the scientific debate on organizational tensions, which has examined an increasingly broad range of contradictions faced by organizations. These authors suggested that there are two main alternative perspectives on the nature and development of tension in organizations, going on to propose a possible integrated model.

The growing number of organizational contradictions that have recently been investigated by scholars include the traditional tension between exploitation and exploration [46–51], as well as the tensions between competition and cooperation [52,53], structure and agency [54–56], and organizations' private and social missions [57–59]. Especially salient to the analytical perspective brought to bear in our own study is the tension between designed versus emergent structures [60–62].

## 5. Conclusions

In analyzing this complex organizational change project across three industrial plants in two different countries, we identified several hundred conceptual codes and produced dozens of categorical maps, ultimately building up the overall representation of the change process just outlined. The change leader plays a key role in the coordination of this process, interacting with multiple actors, and managing people, project, and technical issues, while balancing the strategic objectives defined at the corporate level with salient contextual factors. Paradoxically, the most important concept that emerged from our study, which is uncertainty, was never explicitly mentioned by any of our informants. Nevertheless, this uncertainty took the form of tensions between competing extremes (manual vs. automated, planned vs. emergent, and corporate vs. site), which, in a way that reflected the different perceptions and narratives of the actors, conditioned the course—and final outcomes—of the change process. Managing such tensions implies overseeing processes of negotiation among the different actors, giving rise to a complex dynamic process that must be coordinated by the industrial engineering function, newly transformed from the status of technical staff to that of an internal business partner, as borne out by the tasks now assigned to it and the new competencies that it has developed.

Informed by an existing perspective that sees organizational change as tied up with the management of uncertainty, our analysis shows that uncertainty manifests itself in terms of tensions between competing opposite poles. This opens a series of possible avenues for future inquiry: for example, into the dynamics of these tensions and the factors underpinning their effective management, including in terms of the role of organizational learning in reducing uncertainty and producing corporate knowledge.

From a managerial point of view, this study suggests two key points: first, effective leadership of change relies on the conscious acceptance of uncertainty as well as of the different perspectives and narratives of the actors involved in the change process. Second, it is crucial to effectively recognize and manage tensions between opposite options in relation to key decisions surrounding the desired change.

## 6. Limitations and Future Research

An intrinsic limitation of this study is that it was based on an in-depth analysis of a single global organization. The different views and perspectives given here by the comparison of different industrial sites in more European countries can be further enhanced by additional investigations carried out in multiple organizations, aimed at extending and generalizing the current findings.

The detection and recognition of tensions stemming from uncertainty in organizational change disclose promising areas of further research: first, a specific analysis of tensions, their nature, and their effects on change processes; second, a dynamic model of the process of tension management, and of its interrelations with uncertainty, organizational knowledge, and organizational learning; and third, the investigation of actual decisions and actions implemented in the field for tension management, and the analysis of tension management criteria and critical factors.

**Author Contributions:** F.V. and C.G. contributed equally in all phases of the research project, including conceptualization, methodology, data gathering, analysis, validation, writing, review and editing. Conventionally, half of the content should be attributed to F.V. (Sections 1, 2 and 4–6); the other half of the content (Section 3) should be attributed to C.G. Both authors have read and agreed to the published version of the manuscript.

**Funding:** This research was supported by the Department of Human Sciences for Education, University of Milan-Bicocca, and by the Department of Economics and Business, University of Sassari: (1) fondo di Ateneo per la ricerca 2020; (2) "Dipartimenti di Eccellenza" Program (2018–2022), Italian Ministry of Education (MIUR).

**Informed Consent Statement:** Not applicable.

**Data Availability Statement:** More information and eventually anonymous excerpts of data presented in this study are available on request from the corresponding author. The full data are not publicly available as they would not be fully anonymous.

**Acknowledgments:** Preliminary outcomes of this research at an earlier stage of development were presented at ItAIS 2017, the 14th conference of the Italian Chapter of AIS (Association for Information Systems). The authors would like to acknowledge conference reviewers and participants for useful feedback.

**Conflicts of Interest:** The authors declare no conflict of interest.

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
