# Peer review of "Uncertainty and Emerging Tensions in Organizational Change: A Grounded Theory Study on the Orchestrating Role of the Change Leader"

_sustainability, doi:10.3390/su13094776_

Round 1
Reviewer 1 Report
- The topic is interesting and the grounded theory is suitable for analysis.
- The authors emphasize change management but in the same time the importance of IE in implementing change, creating thus the question if the paper is about change or about IE? It is normal to discuss about IE as a change agent but there are too many places where the authors concentrate their attention on the importance of IE.
- Figures 5-13 presenting conceptual maps are not clear from readability point of view. There are too many words and the fonts are minuscule, such that reading them is almost impossible. The authors must redesign these figures because they are useless in the actual format. As a suggestion, just reduce the number of words and the number of elements such that you can increase the fonts and figures can be understood.
- Once the grounded theory is explained in the methodology, the authors must avoid repeating the codification categories all the time because the effect is to lower the coherence of the presentation logic. It is very difficult to understand what it is actually the point the authors want to make due to this broken discourse.
- The authors use the expressions "critical vision" and "realistic vision" in Figure 4, p. 7. Both of them are inappropriate with respect to the semantic of the concept "vision". The authors should avoid these kind of contributions.
- The authors discuss about a "paradox perspective" citing a paper by Hargrave and Van de Ven. In this kind of change where the degree of complexity is rather law, there is no need to introduce such a discourse about paradoxes.
- The authors discuss about "tensions" like about some kind of discovery. However, a change process is always a battle between forces promoting change and forces apposing change. Thus, tensions are inherent in any change process in a quite natural way.
- The authors should explain more about the resistances specific these changes and to let apart big words and any philosophical perspective that is not necessary for such a change process.
Author Response
Response to Reviewer 1 Comments
- The topic is interesting and the grounded theory is suitable for analysis.
Response 1: Thank you for your appreciation
- The authors emphasize change management but in the same time the importance of IE in implementing change, creating thus the question if the paper is about change or about IE? It is normal to discuss about IE as a change agent but there are too many places where the authors concentrate their attention on the importance of IE.
Response 2: we reduced the explicit references to Industrial Engineering, also to increase the degree of generalization of results and discussion. In particular, we changed “Industrial Engineering” into “change leader”, consistently with the title and abstract, in the following lines: 31, 63, 297, 311, 315, 332, 481, 579, 585, 591, 599/600, 601, 644.
- Figures 5-13 presenting conceptual maps are not clear from readability point of view. There are too many words and the fonts are minuscule, such that reading them is almost impossible. The authors must redesign these figures because they are useless in the actual format. As a suggestion, just reduce the number of words and the number of elements such that you can increase the fonts and figures can be understood.
Response 3: As suggested also by reviewer 3, the readability problem is particularly evident in figures 6 and 13, given their big size. We split these figures in two parts (6a, 6b, 13a and 13b).
- Once the grounded theory is explained in the methodology, the authors must avoid repeating the codification categories all the time because the effect is to lower the coherence of the presentation logic. It is very difficult to understand what it is actually the point the authors want to make due to this broken discourse.
Response 4: Following this suggestion, we have reduced explicit references to actual codes and categories, keeping just a few of them, to help the reader to keep a clear connection between text and maps, as noticed also by reviewer 3.
- The authors use the expressions "critical vision" and "realistic vision" in Figure 4, p. 7. Both of them are inappropriate with respect to the semantic of the concept "vision". The authors should avoid these kind of contributions.
Response 5: Both the expressions have been now removed from figure 4.
- The authors discuss about a "paradox perspective" citing a paper by Hargrave and Van de Ven. In this kind of change where the degree of complexity is rather low, there is no need to introduce such a discourse about paradoxes.
Response 6: We agree with the reviewer that an analysis focused on the concept of paradox may be immature at this point. We therefore removed the last 18 lines from section 4 where the paradox concept had been introduced, reducing also the redundant philosophical content of the discussion as suggested in the following point 8.
- The authors discuss about "tensions" like about some kind of discovery. However, a change process is always a battle between forces promoting change and forces apposing change. Thus, tensions are inherent in any change process in a quite natural way.
Response 7: Thank you for pointing out the opportunity to distinguish between tensions and resistance to change. We added the following sentence in Section 4, lines 595 and followings:
“Tensions may often be related to the well-known phenomenon of resistance to change, but the nature and the sources of tension are well beyond resistance and are related to the wider areas of uncertainty characterizing organizational change”.
- The authors should explain more about the resistances specific these changes and to let apart big words and any philosophical perspective that is not necessary for such a change process.
Response 8: Thank you for this suggestion, we addressed the problem of resistance to change in response 7, and we significantly reduced the philosophical part of discussion focused on the concept of paradox with response 6:
Reviewer 2 Report
Thank you for reading this interesting article. Please consider adding a "Limitations and future research" section.
Author Response
Response to Reviewer 2 Comments
- Thank you for reading this interesting article. Please consider adding a "Limitations and future research" section.
Response 1: Thank you for your appreciations and your suggestion. We added a final section 6 with limitations and future research
Reviewer 3 Report
Review
The paper presents an impressive investigation to change management in the parcel delivery industry using a solid Grounded Theory approach.
Methods
Some small issues:
- The part on onsite visits (2.2) and framing (2.3) needs some reorganisation and clarifications. When reading the section 2.2. the third visit seems to be missing. This visit is later found in section 2.3. Suggestion: push the part on the third visit, starting on line 174, to the end of section 2.2.
- Suggestion for section 2.2: remove the narrative stance in the section, i.e., “the next day…”. This adds nothing to description.
- Section 2.3. Please add a sentence (explanation) after the theoretical concept of data saturation.
Results
- The number of the headline 3.3 should be altered to just 3 (line 182).
- The figures add to the understanding of the results. However, figure 6 and figure 13 is quite hard to read because of the size of the figure. Something to consider, maybe?
References
- Armenakis and Harris (2009) are mentioned twice in the reference section. Delete one entry.
Reviewer 4 Report
The paper clearly states the knowledge of the authors regarding the theme under investigation, nevertheless, there are some areas that need further improvement to accentuate the academic nature of the article. Please check the following areas:
Abstract
Please ensure that this part clearly refers to the following topics:
Purpose;
Design;
Methodology used;
Findings;
Practical implications;
Originality/value.
Introduction
The introduction should be supported as much as possible by literature, at the end of this part add a short description of how the paper is organised.
Literature Review
Add a chapter with the relevant literature
Methodology
The methodology should be revised and reinforce, start this chapter by detailing the type of methodology to be used (supported by literature).
Hypothesis application could benefit the methodology
Discussion of results
The results need to be deepened to allow a discussion supported by literature
Conclusions
Please revised this part, ensuring that the following topics are clearly described and supported as much as possible by relevant literature:
- Main findings;
- Theoretical contributions;
- Practical contributions;
- Research originality;
- Research limitations;
- Future lines of investigation.
Round 2
Reviewer 1 Report
The authors performed the recommended modifications in the initial manuscript. The revised version is much better.